



# Effects of Nonmigrating Diurnal Tides on the Na Layer in the Mesosphere and Lower Thermosphere

Jianfei Wu[1,2,3], Wuhu Feng[4,5], Xianghui Xue[1,2,3,6], Daniel R. Marsh[4], and John M. C. Plane[4]

[1]CAS Key Laboratory of Geospace Environment, School of Earth and Space Sciences, University of Science and Technology of China, Hefei, China
[2]Mengcheng National Geophysical Observatory, School of Earth and Space Sciences, University of Science and Technology of China, Hefei, China
[3]CAS Center for Excellence in Comparative Planetology, China
[4]School of Chemistry, University of Leeds, Leeds, UK
[5]National Centre for Atmospheric Science, University of Leeds, Leeds, UK
[6]Collaborative Innovation Center of Astronautical Science and Technology, 230026, Hefei, China

**Correspondence:** Xianghui Xue (xuexh@ustc.edu.cn)

**Abstract.** The influence of nonmigrating diurnal tides on Na layer variability in the mesosphere and lower thermosphere regions is investigated for the first time using data from the Optical Spectrograph and InfraRed Imaging System (OSIRIS) on the Odin satellite and Specified-Dynamic Whole Atmosphere Community Climate Model (SD-WACCM) with metal chemistry. The Na density from OSIRIS exhibits a clear longitudinal variation indicative of the presence of tidal components. Similar variability is seen in the SD-WACCM result. Analysis shows a significant relationship between the nonmigrating diurnal tides in Na density and the corresponding temperature tidal signal. Below 90 km, the three nonmigrating diurnal tidal components in Na density show a significant positive correlation with the temperature tides. Conversely, the phase mainly indicates a negative correlation above 95 km. Around the metal layer peak, the response of the Na density to a 1 K change in tidal temperature is estimated to be 120 cm$^{-3}$.

## 1 Introduction

In the mesosphere and lower thermosphere (MLT), the atmospheric structure is significantly influenced by atmospheric tides with different spatial and temporal scale variations. Meteor-ablated metals such as sodium (Na), iron (Fe) and potassium (K) can be characterized as the optimal dynamical trace medium in the MLT (80-120 km in altitude), because their removal lifetimes are relatively long (Plane et al., 2015). The diurnal cycle of these meteoric atoms is driven by a complex combination of tidal forcing, chemistry and photochemistry in this region. Although several recent studies (e.g., Feng et al., 2013, 2015; Viehl et al., 2016; Yuan et al., 2019) have investigated the diurnal variation of the Na and Fe layers using a whole atmosphere chemistry-climate model, their findings mainly focused on photochemical influences. The mechanism of the diurnal variation still needs to be fully understood.

Nonmigrating tides exhibit stable wave number structures that do not propagate with the sun, resulting in relatively fixed longitudinal patterns. Thus, the study of non-migrating tides is crucial for understanding the mechanisms of energy, chemical,



and dynamical transport within the MLT region (Oberheide and Forbes, 2008; Smith, 2012; Ramesh and Smith, 2021; Li et al., 2022). Longitudinal variations observed from sun-synchronous orbit satellites, which provide data at consistent local times, enable the removal of solar illumination and photolysis effects, thereby allowing for a focused analysis on non-migrating diurnal variations (Oberheide and Forbes, 2008). This approach facilitates a clearer understanding of the stable tidal structures

that modulate the MLT, distinguishing them from other factors such as solar-driven photochemistry.

Exploiting the accumulation of continuous lidar observations, researchers have shown an increased interest in the diurnal variations of these metals and their possible driving mechanisms in recent decades. Clemesha et al. (1982, 2002) found that the diurnal variation of Na density was related to tidal winds at São José dos Campos (23°S, 46°W). In contrast, further analysis found that the diurnal variation in Na was largely influenced by photochemistry rather than tidal forcing on the bottom side of

the Na layer (States and Gardner, 1999; Yuan et al., 2012, 2019). Comparisons of lidar observations from the Chinese Meridian Project indicate that there are considerable regional differences in the Na diurnal cycle across China. The diurnal variations appear to be principally driven by photochemistry over Beijing (40.5°N, 116°E), while tidal modulations dominate over Wuhan (30.5°N, 114.6°E) (Gong et al., 2015; Xia et al., 2020a, b). These observations indicate that the mechanisms controlling the diurnal variability of the Na layer are complex. Due to the constraints of point observations, and the interplay between the

tides, solar-driven photochemistry and atmospheric background conditions, it is not possible to identify a single mechanism.

In this study, we investigate the response of the diurnal variations of atomic Na to nonmigrating diurnal tides using output from the Specified Dynamics (SD) Whole Atmosphere Community Climate Model (WACCM), which is nudged by NASA's Modern-Era Retrospective Analysis for Research and Applications, Version 2 (MERRA-2) product (Molod et al., 2015). Owing to the characteristics of its sun-synchronous orbit, the Na density retrieved from the Optical Spectrograph and InfraRed

Imaging System (OSIRIS) on the Odin satellite exhibits clear longitude variations, indicating the presence of tidal components. Specifically, the correlation of the temperature nonmigrating diurnal tide and metal layer variability provides evidence for a mechanism for how temperature modulates the metal layer variation. In section 2, we describe the model and observational data in more detail. The results of the analysis are presented in section 3, and the conclusions follow in section 4.

## 2   Model Description and Data Analysis

### 2.1   Model Description

Our results are derived from a simulation of SD-WACCM (Smith et al., 2017; Yuan et al., 2019), a modified version of WACCM6 nudged to meteorological fields from the MERRA-2 analysis. WACCM6 is a high-top atmospheric model within the framework of the Community Earth System Model Version 2 (Gettelman et al., 2019), with 88 vertical levels up to approximately 140 km altitude. The horizontal resolution of this simulation is $1.9° \times 2.5°$, and the time interval of the output data is

1 hr. This setup also includes the validated metal chemistry module for Na (Marsh et al., 2013) with updated rate coefficients from Plane et al. (2015).

WACCM is widely used in atmospheric tidal studies in the MLT region and compares satisfactorily with various observations (e.g., Chang et al., 2008; Lu et al., 2011, 2012; Davis et al., 2013; Ramesh and Smith, 2021). The WACCM model with Na,



Fe, and K chemistry schemes has been successfully used to simulate the periodic characteristics of the MLT metal layers. For
example, the teleconnections between metal layer variations and the solar cycle have been investigated by the model output and
confirmed by parallel satellite observations (Dawkins et al., 2016; Wu et al., 2019). The simulated diurnal cycle of Fe, Na and
K in WACCM compares favourably with lidar observations, except that the peak height of the modelled layers is 2-3 km lower
than observations (Feng et al., 2015; Viehl et al., 2016; Yuan et al., 2019), which may be due to slow vertical transport caused
by the unresolved waves in the gravity parametrization scheme (Yu et al., 2022). Due to the ability of this state-of-the-art model
to simulate periodic diurnal changes in the metal layer close to observations, it is possible to investigate the response of the
metal layers to nonmigrating diurnal tides.

## 2.2   Data Analysis

The Na layer observations measurements in the equatorial region (defined hereafter as 15°S-15°N) that are used here come
from nearly ten years of Na number density measurements made by OSIRIS/Odin during 2004–2013 (Langowski et al., 2017).
Odin is a limb-scanning satellite in a sun-synchronous orbit, with an ascending node at about 1800 local time (LT) and a
descending node at about 0600 LT. The Na D-lines at 589 nm in the dayglow emission are predominantly produced by solar-
stimulated resonance fluorescence of Na atoms. The limb-scanning measurements of the D-line emission, made by the OSIRIS
spectrometer, are then converted into absolute Na atomic concentration profiles using the retrieval scheme of Gumbel et al.
(2007). Hence, the Na observations are daytime only (∼0600 and 1800 LT), which removes a potential photochemical day/night
variation when analysing the diurnal tidal signal. Due to the severe lack of adequate data at the ascending node (1800 LT) over
the equatorial latitudes, we mainly present the observation results of the descending node (0600 LT).

Using SD-WACCM data from 2004 to 2013, with a 1-hour time resolution, we extracted the amplitudes and phases of tides
through a two-dimensional Fast Fourier Transform (2D FFT). The method is applied to the time-longitude series of Na, U, V,
W, and T at each latitude and height level. In this paper, we limited our analysis to the three dominant tidal components: DW2
(Diurnal Westward Wavenumber 2), DE3 (Diurnal Eastward Wavenumber 3), and DS0 (Diurnal Stationary Tide), because there
is much observational evidence showing that these are the most prominent components of the nonmigrating diurnal tides in the
MLT region (Forbes et al., 2003; Wu et al., 2008; Zhang et al., 2010; Lu et al., 2011).

## 3   Results and Discussion

Figure 1 shows the longitude/altitude distribution of the composite monthly average equatorial Na density from OSIRIS/Odin
observations between 2004 and 2013 at 0600 LT (descending node) and the corresponding WACCM model outputs for the same
period at 0600 LT for March, June, September, and December. The corresponding temperatures from WACCM are presented
with contour lines in the right panels. As illustrated in Figure 1, the WACCM Na peak density (∼4000 cm$^{-3}$) is approximately
30% higher than the observations, and the peak height (∼ 90 km) is 2-3 km lower, which is consistent with the results reported
in the previous studies (Feng et al., 2015; Viehl et al., 2016; Yuan et al., 2019; Yu et al., 2022). The Na density distributions





from both the OSIRIS/Odin observations and WACCM exhibit a pronounced longitudinal variation. The Na number density
from OSIRIS/Odin and WACCM is much lower in March than in other months (June, September, and December).

Although extracting tidal signals at a fixed local time is challenging, the total coverage at the same local time provides
an excellent opportunity to show the longitudinal variations influenced by tidal components. In the left panels of Figure 1,
the OSIRIS/Odin observations exhibit an apparent wave-3/wave-4 longitudinal structure in June, while there are only single or

double maxima in other months. In the right panels of Figure 1, the WACCM model data show visible wave-2 or wave-3 patterns
in June and September, and the other months exhibit more hybrid structures. It should be noted that the WACCM simulation
and Odin observations are not expected to match precisely due to inherent differences in model assumptions, data resolution,
and the specific physical processes represented. The longitudinal variation of Na density at the peak height ($\sim 90$ km) in the
WACCM is consistent with the temperature variation (contour lines) at the corresponding height (right panels of Figure 1).

For example, each density maximum corresponds precisely to the temperature maximum in June. Specifically, in June, the Na
density peaks at around 4000 cm$^{-3}$, coinciding with temperature maxima of approximately 190 K. This longitudinal variation
is suggested to be attributed to the tidal temperatures, where Na density is positively correlated with temperature (Plane et al.,
1999, 2015; Wu et al., 2019). We also investigated the role of tidal winds (zonal and meridional winds, as well as vertical
transport). We found no statistically significant relationship between these tidal winds and Na density (not shown).

Figure 2 shows the equatorial Na column density residuals at 0600 LT from the WACCM outputs (blue line) and OSIRIS/Odin
observations (orange line), respectively. The residuals are obtained by subtracting the zonal mean data from the monthly mean
column density profiles, which eliminates migrating tidal impacts and retains only perturbations due to nonmigrating tides.
Although the Na peak heights and densities of the two datasets are different (Figure 1), their column density residuals are
of the same order. Notably, the range of Na column density residuals (difference between the highest and lowest values) is

$\sim 1.0 \times 10^9$ cm$^{-2}$, indicating similar magnitudes of nonmigrating tidal influences in both datasets. The significant longitudinal
variation is further supported in the Na column density residuals, and is also consistent with the Na concentration variations
with longitude shown in Figure 1. In some months, the Na column density residuals from Odin and WACCM overlap signifi-
cantly (e.g., February, March, July, and September), while in other months, they exhibit phase differences (e.g., May, August,
November, and December). However, both distributions consistently display patterns of superimposed wave oscillations. As

expected, the disturbances are particularly intense from January to February and from May to October, coinciding with the
significant presence of nonmigrating tidal components (DW2 and DE3) during these months.

Since Odin Na measurements were essentially limited to the same local time, this provides a rare opportunity to measure
the longitudinal variations introduced by nonmigrating tides. However, extracting the relevant fluctuations in amplitude is
challenging due to this observational limitation. Consequently, we will use the WACCM model results for quantitative analysis.

Furthermore, SD-WACCM, constrained by MERRA-2 data below 60 km, is well-suited for investigating the mechanisms of
tidal influence on the sodium layer.

Figure 3 shows the seasonal variation of the tidal amplitudes and phases of DW2, DE3 and DS0 for Na number density from
WACCM over the equatorial region, with superposed contour lines of corresponding temperature tides. There are two peaks
in the maximum Na density tidal amplitudes (left column in Figure 3). The primary peak amplitude of the Na density tidal





**Figure 1.** Composite monthly average Na density from OSIRIS/Odin observations (left), WACCM model output (right) over the equatorial region for March, June, September, and December. The Odin observations presented here were from the descending part of the orbit at 0600 LT for the years 2004 to 2013, and the WACCM results were for the same period at 0600 LT. Contour lines on right panels denote WACCM temperatures.



**Figure 2.** Composite monthly average equatorial Na column density residuals from Odin (orange line) and WACCM (blue line) with standard error bars for both datasets from 2004 to 2013.

variations ranges from 400 cm$^{-3}$ to 500 cm$^{-3}$, occurring at 80-90 km, while the second peak altitude is ∼10 km higher with an amplitude that is about half that of the first peak. The magnitude of the DW2 and DE3 tidal amplitudes is comparable, but much stronger than those of DS0, which is consistent with their temperature tidal amplitudes at the corresponding heights. DW2 and DE3 exhibit peak amplitudes up to 500 cm$^{-3}$, whereas DS0 typically exhibits peak amplitudes around 350 cm$^{-3}$. The seasonal distribution of the maxima/minima in the magnitude of the Na tidal amplitudes is generally consistent with the temperature component at the corresponding altitude, and the seasonal variations of the amplitude of the two vertical peaks are roughly synchronous. Generally, the response of the Na density to a 1 K change in tidal temperature is estimated to be 120 cm$^{-3}$ at the primary peak height (∼87 km). As shown in Figure 3, DW2 is comparable in most seasons, with the maximum occurring from September to November and the minimum in May; DE3 is strongest during June-October and less strong in





April; and DS0 is prominent in July. This seasonal variation pattern is consistent with that of Forbes et al. (2003). Regarding
the phase of Na density with that of temperature, their correlation varies with altitude, with a critical level of ∼90 km.



**Figure 3.** Seasonal variation of three nonmigrating diurnal tidal components of Na number density (color shading) and temperature (contours)
for amplitudes (left panels) and phases (right panels) over the equatorial latitudes as a function of month and altitude.

The Pearson correlation coefficient distribution between the monthly variations of Na number density and those of temper-
ature at the equatorial latitudes is presented in Figure 4 for these three components. Between 80 and 92 km, the three tidal
components share a common feature: both amplitude and phase in Na density exhibit a significant positive correlation with




temperature tides. This indicates that increases in temperature are generally accompanied by increases in Na density. In con-
trast, above 95 km, the Na tidal amplitude continues to be positively correlated with temperature, suggesting that temperature
fluctuations still drive changes in Na density. However, the phase shows a marked negative correlation, indicating a phase shift
where increases in temperature correspond to decreases in the Na density. This shift highlights a complex interaction between
Na density and temperature at higher altitudes.

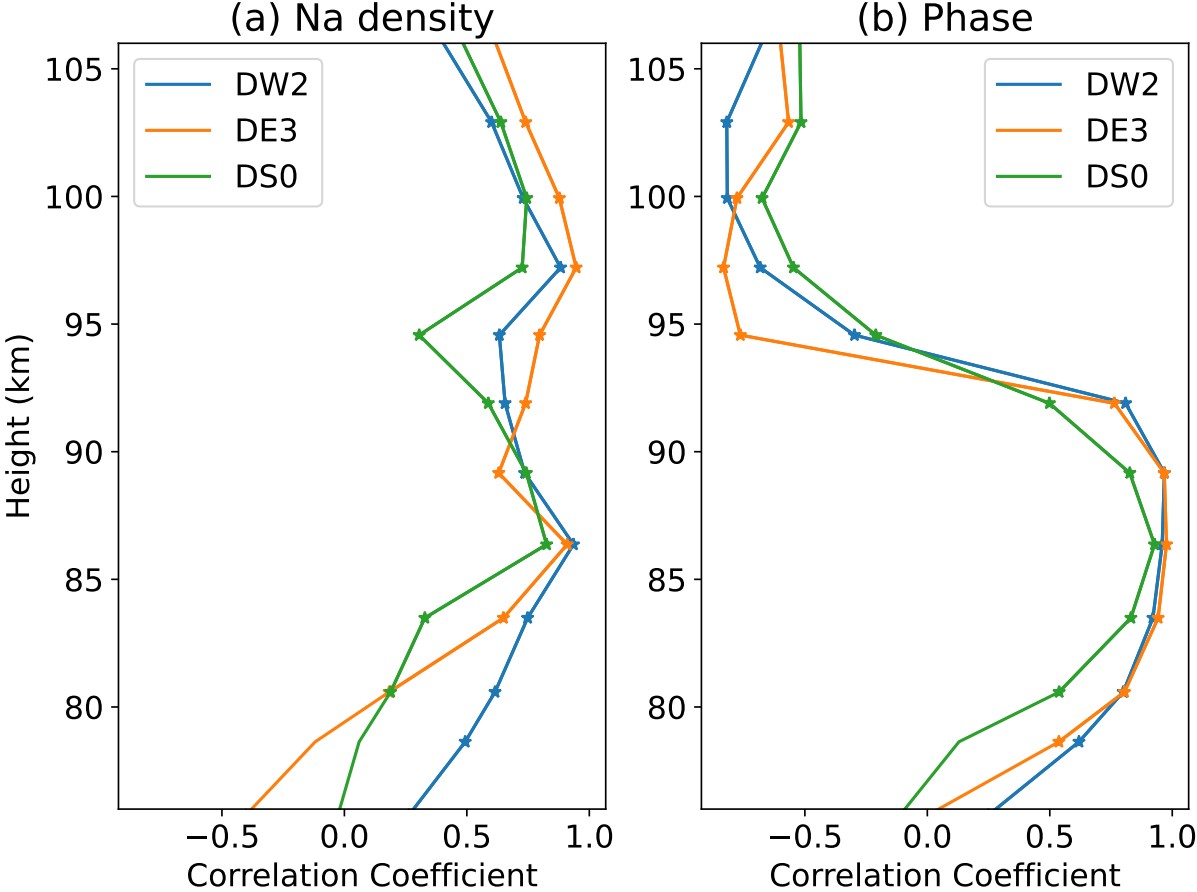

**Figure 4.** The Pearson correlation coefficient between Na number density and temperature variations for amplitudes (a) and phases (b) at the
equatorial latitudes as a function of altitude (76–110 km). Those at 95% confidence levels are marked with asterisks.

The altitude-dependent correlation of Na number density variations to temperature variations seems consistent with the
study, which found that Na exhibits a positive correlation with the simulated temperature at approximately 90 km (Figure 10 in
Plane et al. (1999), and Figure 2 in Wu et al. (2019)). Plane et al. (1999, 2015) provide a possible explanation for the correlation
between metal atoms and temperature in the MLT region, and herein we give a brief summary. Over the height range ~80-90
km, neutral chemistry dominates. The metal Na atoms are released from the reservoir $NaHCO_3$ by reaction with atomic H
($NaHCO_3 + H \rightarrow Na + H_2CO_3$); the rate coefficient has a positive activation energy (Plane et al., 2015) and so increases at





higher temperatures. The explanation for the second peak altitude at $\sim 95$ km is more complicated. The Na reservoir species here is $Na^+$. This is converted back to Na by recombination with $N_2$, followed by switching with $CO_2$ to form the $Na^+.CO_2$ which then undergoes dissociative electron recombination (Cox and Plane, 1998). The recombination reaction has a negative temperature dependence, and so is slower at higher temperature. Moreover, atomic O can disrupt the neutralization sequence by reacting with $Na^+.N_2$ to form $NaO^+$, which then reacts with a second O to reform $Na^+$. After examining the relationship

between atomic O and temperature, we found that atomic O is positively correlated with temperature (not shown). Thus, Na should be less strongly correlated with temperature on the top-side of the layer (Plane et al., 1999). Indeed, Figure 4(b) shows that the Na phases of all three tidal components exhibit a significant anticorrelation with the temperature phase between 95 and 100 km, which is consistent with the anti-correlation of Na density and temperature above 96 km from pretvious lidar measurements and numerical simulations (Plane et al., 1999).

Figure 5 illustrates the vertical profiles of the mean nonmigrating diurnal variation and diurnal variation from 2004-2013 at equatorial latitudes ($\pm 15°$), and includes the mean Na density profile (left panel). It also shows their percentages relative to the mean Na density (right panel). The diurnal variation reaches its peak of around 900 cm $^{-3}$ at approximately 90 km, aligning with the peak height of the Na density. In contrast, the nonmigrating diurnal variation has a mean peak amplitude of 350 cm$^{-3}$, occurring at a slightly lower altitude of $\sim 87$ km, which is consistent with the findings shown in Figure 3. This

peak amplitude of the nonmigrating diurnal tides in Na density represents $\sim 10\%$ of the mean Na peak density (3500 cm $^{-3}$) and contribute $\sim 40\%$ of the total diurnal variation in Na density. In this region, the remaining diurnal Na density variations may be due to the combined effects of migrating diurnal tides, photochemistry and non-tidal effects. More full diurnal cycle observations are required to investigate their contributions (States and Gardner, 1999; Yuan et al., 2012; Xia et al., 2020b). Above 95 km, the contributions of the nonmigrating diurnal tidal amplitude increases gradually to a maximum around 100 km

of 25%, corresponding to the second peak height result shown in Figure 3. Interestingly, on the bottom side of the Na layer (around 80 km), where photochemistry plays a major role, it can be inferred from the two percentages that the Na diurnal variation is comparable to the Na density, which is in good agreement with lidar observations (States and Gardner, 1999; Yuan et al., 2019; Xia et al., 2020b).

## 4   Conclusions

The nonmigrating diurnal tidal variation of the metal layers in the MLT region has been investigated using data from OSIRIS/Odin observations and the SD-WACCM model with added metal chemistry. Whereas previous studies focused on the influence of photochemistry on the diurnal variations in Na density on the top and bottom sides of the layer, to the best of our knowledge this is the first time that the nonmigrating diurnal tidal signal of a metal layer has been investigated. The present study found the following features: (1) the Na density exhibits longitudinal variations, indicating tidal components in both datasets, although

there are differences between the two datasets which vary month-to-month; (2) there is a significant relationship between the nonmigrating diurnal variation in Na density and the corresponding temperature tidal signal, rather than with dynamics such as the wind tides; (3) the mean amplitude of the nonmigrating diurnal variation in Na density at the equatorial latitudes is $\sim 350$



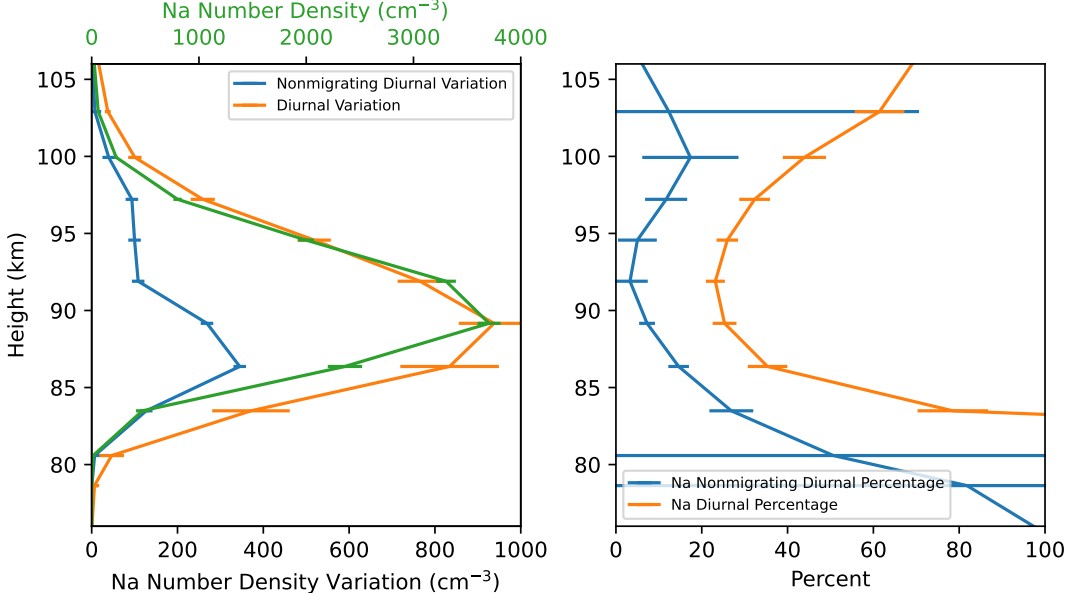

**Figure 5.** Vertical profiles of the 2004-2013 mean nonmigrating diurnal variation (blue) and diurnal variation (orange) at equatorial latitudes (left panel), and their percentages relative to the mean Na density (right panel). The green profile represents the mean Na density.

cm$^{-3}$ at 87 km, accounting for 10% of the mean Na number density and contributing ∼40% of the total diurnal variation of Na; (4) below 90 km, the amplitude and phase of the three nonmigrating diurnal tidal components in Na density show a significant
positive correlation with the temperature tides, while above 95 km, the phase shows a negative correlation.

Previous research has established the diurnal variation on the top and bottom sides of three metal layers (Na, K, and Fe) by lidar and modelling (e.g., States and Gardner, 1999; Yuan et al., 2012; Yu et al., 2012; Feng et al., 2015; Yuan et al., 2019; Xia et al., 2020a). This study is the first to extract nonmigrating diurnal tidal signals from diurnal variations through a global atmospheric metal layer model, and a significant magnitude is found around the Na layer peak. In the near future, the
construction and continuous observation using high-precision lidars at the same latitude, which is part of the Chinese of the Meridian Project Phase II, will provide new opportunities to investigate and understand nonmigrating tidal variations in the metal layers.

*Code and data availability.*  The SD-WACCM model output data presented in this study is available at Wu et al. (2024). The WACCM6 model code can be accessed from https://www.cesm.ucar.edu/working-groups/whole-atmosphere. The derived OSIRIS/ODIN Na data (Dawkins
et al., 2016) presented in this study is available at Wu et al. (2023).



*Author contributions.* J. W. in close collaboration with W. F., J. M. C. P. and D. R. M. designed the experiment, ran the WACCM model, J. W. performed the data analysis, and prepared the manuscript. W. F., X.X., and J. M. C. P. contributed to the interpretation of results and the writing of the manuscript.

*Competing interests.* At least one of the (co-)authors is a member of the editorial board of Atmospheric Chemistry and Physics.

*Acknowledgements.* This work was supported by the National Key R&D Program of China (Grant No. 2022YFF0503703), the National Natural Science Foundation of China (42125402, 42188101, 42374183, and 42074181), the Project of Stable Support for Youth Team in Basic Research Field, CAS (YSBR-018). J. W. was funded by the foundation of the Fundamental Research Funds for the Central Universities (Grant No. WK3420000020). W. F., J. M. C. P. and D. R. M. are supported by the jointly funded project by the USA National Science Foundation's Directorate of Geosciences (NSFGEO) and the UK National Environment Research Council (grant no. NE/T006749/1). W. F.
is partly funded by NCAS Visiting Scientist Programme. The numerical calculations in this paper were undertaken on the supercomputing system in the Supercomputing Center of the University of Science and Technology of China.



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
