# Peer review of "Effects of Nonmigrating Diurnal Tides on the Na Layer in the Mesosphere and Lower Thermosphere"

_EGUsphere, 2024_

## Author Comment (AC1)

We thank the reviewers for their detailed, helpful, and overall supportive comments. We have revised the manuscript to account for each comment. Responses to the individual comments are provided below. Reviewer comments are in black. Author responses are in blue. Line numbers in the response correspond to those in the revised manuscript, and the changes are shown in the version of the manuscript with track changes.

**Reviewer #1:**

The manuscript " Effects of Nonmigrating Diurnal Tides on the Na Layer in the Mesosphere and Lower Thermosphere", by Wu et al., analyze the variability of the Na layer in the mesosphere and lower thermosphere using experimental as well as modeled data from the Optical Spectrograph and InfraRed Imaging System (OSIRIS) and from the Specified-Dynamic Whole Atmosphere Community Climate Model (SD-WACCM) with metal chemistry, respectively.

The paper is well-structured, the methodology is well explained, and the authors made their data publicly available, both the experimental data and those obtained from the model simulations.

A novelty of the present work is its focus on the nonmigrating tide signal, and its assessment from diurnal variations through a global atmospheric metal layer model. The results are clearly presented and they address a relevant topic within the scope of ACP journal.

I consider it can be accepted for publication in its present form and I only have one comment and a "technical" correction.

My comment is the following:

Regarding Figure 3, how stable are the amplitude and phase values along the period you considered (2004-2013) ? I mean, does it vary with the solar activity level for example. Since this period corresponds to almost a full cycle (from the maximum of solar cycle 23 to the maximum of cycle 24, including the deep 2008 minimum). I guess, anyway, this does not affect your results. I leave to the authors to decide whether or not to explain some of this variability, if they feel it could add something extra to this very good work.

**Response:** Thank you for this very useful comment. As you point out, our data (2004-2013) span nearly a full solar cycle, covering both the maximum of solar cycle 23, the deep solar minimum of 2008, and the rise to solar cycle 24. To address your suggestion, we investigated the potential influence of solar activity on the nonmigrating tidal components in the Na layer. We performed a comparative analysis, treating 2007-2009 as the solar minimum years and 2004-2006 and 2011-2013 as solar maximum years. The results, as shown in Figures R1-R2, indicate that solar activity does not significantly impact the amplitude and phase values of the nonmigrating tidal components, which is consistent with your expectation.

**Changes to Manuscript**: "Since the data covered almost a full solar cycle (2004-2013), our investigation extended to the impact of solar activity on the seasonal distribution of these

nonmigrating tidal components and found that solar activity had no significant impact on the results (not shown)."

**Changes:** Please see Lines 134-137.

[Figure]

**Figure R1.** Seasonal variation of three nonmigrating diurnal tidal components of Na number density (color shading) and temperature (contours) for amplitudes (left panels) and phases (right panels) over the equatorial latitudes as a function of month and altitude during the 2007–2009 solar minimum.

[Figure]

**Figure R2.** Seasonal variation of three nonmigrating diurnal tidal components of Na number density (color shading) and temperature (contours) for amplitudes (left panels) and phases (right panels) over the equatorial latitudes as a function of month and altitude during the 2004–2006 and 2011–2013 solar maximum periods.

Technical correction:

Figure 3: Please, check the colorbar values of the Phase plots. I think it should be from -12 to 12.

**Response:** Thank you for pointing this out. In the revised manuscript, we have corrected the colorbar on the left side of Figure 3.

**Reviewer #2:**

The paper studied the mechanism of the diurnal variation of meal layer by analyzing observation from the Optical Spectrograph and InfraRed Imaging System (OSIRIS) and model data from the Specified-Dynamic Whole Atmosphere Community Climate Model (SD-WACCM) with metal chemistry.

The paper is well written, and contributes evidence for a mechanism for how temperature modulates the metal layer variation, which enables distinguishing stable tides from other factors such as solar-driven photochemistry.

There are some problems, which must be solved before it is considered publication. If the following problems are well-addressed, I believes that the essential contribution of this paper are important for the dynamical mechanisms of the MLT region.

My comments are as follows:

The significance of this paper is not expound sufficiently. The author need to highlight this paper's innovative contributions. Relevant research background needs to be supplemented in Introduction.

**Response:** Thank you for your insightful suggestion. In the revised manuscript, we have expanded the Introduction section to better explain the research objectives and provide more detailed background information relevant to this study. We have also emphasized the innovative contributions of this paper, which include a comprehensive analysis of nonmigrating diurnal tidal components in the Na layer. Our study utilized global satellite data and forward modeling to investigate the influence of nonmigrating tides on Na layer fluctuations in the MLT region and to explore their possible pathways. This study provides new insights into the seasonal variations of nonmigrating diurnal tidal components in the Na layer, focusing on how temperature tides influence the distribution of Na.

**Changes to Manuscript**: Expanded the Introduction section, highlighting the innovative contributions, focusing on nonmigrating tides in the Na layer.

**Changes:** Please refer to the Introduction section in the manuscript with track changes.

**Figure 1** I'm confused with Figure 1 and its description. The right column of Figure 1 shows temperature information on May, June, September and December. But the bar title was named "WACCM Na density [cm $^3$]". Also, the "cm $^3$" is wrong, should be "cm$^{-3}$". Please check all the units.

**Response:** The contour lines overlaid on the right column of Figure 1 represent temperature, while the color contours and the color bar correspond to Na density. The color bar title is "WACCM Na

density [cm⁻³]." In the revised manuscript, we have ensured that the font style is consistent, which has improved the overall visual quality of the figure and ensured that the unit is correctly displayed as "cm⁻³."

**Figure 2** "column density" should be changed to "column abundance". And the corresponding unit should be "Na column abundance $(cm^{-2})$".

**Response:** Thank you for pointing this out. In the revised manuscript, we have made the suggested changes to Figure 2, replacing "column density" with "column abundance." Additionally, we have updated the corresponding expressions throughout the manuscript to reflect this change.